# The Novel Rotor Flux Estimation Scheme Based on the Induction Motor Mathematical Model Including Rotor Deep-Bar Effect

**Grzegorz Utrata [1],\*, Jaroslaw Rolek [2] and Andrzej Kaplon [2]**

[1] Institute of Environmental Engineering, Czestochowa University of Technology, Brzeźnicka 60a, 42-200 Czestochowa, Poland

[2] Department of Industrial Electrical Engineering and Automatic Control, Kielce University of Technology, Tysiąclecia Państwa Polskiego 7, 25-314 Kielce, Poland

\* Correspondence: gutrata@is.pcz.pl; Tel.: +48-500-145-449

**Abstract:** During torque transients, rotor electromagnetic parameters of an induction motor (IM) vary due to the rotor deep-bar effect. The accurate representation of rotor electromagnetic parameter variability by an adopted IM mathematical model is crucial for a precise estimation of the rotor flux space vector. An imprecise estimation of the rotor flux phase angle leads to incorrect decoupling of electromagnetic torque control and rotor flux amplitude regulation which in turn, causes deterioration in field-oriented control of IM drives. Variability of rotor electromagnetic parameters resulting from the rotor deep-bar effect can be modeled by the IM mathematical model with rotor multi-loop representation. This paper presents a study leading to define the unique rotor flux space vector on the basis of the IM mathematical model with rotor two-terminal network representation. The novel rotor flux estimation scheme was validated with the laboratory test bench employing the IM of type Sg 132S-4 with two variants of rotor construction: a squirrel-cage rotor and a solid rotor manufactured from magnetic material S235JR. The accuracy verification of the rotor flux estimation was performed in a slip frequency range corresponding to the IM load adjustment range up to 1.30 of the stator rated current. This study proved the correct operation of the developed rotor flux estimation scheme and its robustness against electromagnetic parameter variability resulting from the rotor deep-bar effect in the considered slip frequency range.

**Keywords:** deep-bar effect; mathematical model; estimation; induction motors; motor drives

## 1. Introduction

The development of advanced control methods of induction motors (IMs), such as direct and indirect field-oriented control [1,2] or direct torque control [3], have contributed to the widespread use of this type of motor in modern drive systems intended for various applications in industry. In the rotor-flux-orientation, the stator phase currents through the Park's transformation are represented by the field- and torque-producing components. In cases when the rotor flux amplitude is stabilized by the field-producing component of the stator current space vector, IM electromagnetic torque is linearly proportional to the torque-producing component [1]. The decoupling of IM electromagnetic torque control and rotor flux amplitude regulation is realized based on the phase angle of the rotor flux space vector. Since direct measurement of the rotor flux is practically not achievable, development of indirect methods for rotor flux space vector estimation is reported in the world literature, especially model-based methods.

In IM field-oriented control, slip frequency is controlled within the set range of values, except for very short torque transients. With slip frequency changes, rotor electromagnetic parameters vary due to

the rotor deep-bar effect. For maintaining the high dynamic performance of the IM rotor-flux-oriented control during torque transients, the accurate representation of rotor electromagnetic parameter variability by an adopted IM mathematical model is required.

Inaccurate representation of this variability by the adopted IM mathematical model, which serves as basis for the rotor flux estimation scheme, leads to an erroneous estimation of the rotor flux space vector. In consequence, the erroneous estimation of the vector components results in deterioration of decoupling effectiveness of electromagnetic torque control and rotor flux amplitude regulation, thus deteriorating the overall performance of the IM rotor-flux-oriented control [4–6]. For this reason, the compensation of the influence of the rotor deep-bar effect on the rotor flux estimation accuracy is important for the rotor-flux-oriented control of squirrel-cage IMs, especially the ones where the rotor bar is large enough to incorporate high rotor current.

Until now, estimation schemes for the rotor flux space vector have been elaborated predominantly on the basis of the IM classical mathematical model with rotor single-loop representation with constant parameters. In order to compensate for the influence of the rotor electromagnetic parameter variability on the rotor flux estimation accuracy, the estimation schemes extended by algorithms enabling tracking variability of rotor electromagnetic parameters were proposed [4,6–15]. These schemes work very well with reference to IMs with squirrel-cage rotors, in which the electromagnetic parameters do not show significant variability resulting from the rotor deep-bar effect. The response of the proposed algorithms for variability tracking of rotor electromagnetic parameters may not be fast enough to follow rapid parameter variability during torque transients. These algorithms were mainly intended to model rotor resistance changes associated with temperature variation [4,6–15].

The variability of rotor electromagnetic parameters resulting from the rotor deep-bar effect can be modeled by the IM mathematical model with rotor multi-loop representation [16–25]. Nevertheless, an estimation scheme of the rotor flux space vector which algorithm would be formulated on the basis of such IM mathematical models has not been developed so far. What is more, the authors of these works [19–22] stated that defining the unique rotor flux space vector in the IM mathematical model with rotor multi-loop representation is not possible, and thus they proposed IM airgap-flux or pseudorotor-flux oriented control, developed with the use of the mathematical model of this type.

The results of simulation and experimental studies presented previously [19–22] indicate very good dynamic performance of the vector-controlled squirrel-cage and double-cage IMs. This fact encouraged us to carry on work on the application of the IM mathematical model with rotor multi-loop representation in the IM rotor-flux-oriented control, since such a control strategy has a simpler structure and a more effective decoupling of electromagnetic torque control and rotor flux amplitude regulation than airgap-flux-oriented control [22].

This paper presents a study which leads to development of the rotor flux estimation scheme on the basis of the IM mathematical model with rotor two-terminal network representation. The overall goal of this work was focused on the accuracy verification of the rotor flux estimation in a slip frequency range corresponding to the IM load adjustment range up to 1.30 of the stator rated current. Thus, the considered slip frequency range exceeded the typical operating range of slip frequency for IM field-oriented control. This study aimed to prove the proper modeling of the electromagnetic parameter variability resulting from the rotor deep-bar effect by the novel rotor flux estimation scheme. Due to the assumed concept of the conducted work, the experimental investigations were realized in an open-loop drive system (without speed feedback or slip compensator), at a fixed setpoint of stator voltages and step commands of load torque. The evaluation of operation accuracy of the developed rotor flux estimation scheme was realized indirectly with the use of the registered shaft torque.

The results of the conducted study point out an improvement of the estimation accuracy of the rotor flux space vector obtained by the scheme developed on the basis of the IM mathematical model with rotor two-terminal network representation, in comparison to the accuracy which was gained by the estimation schemes formulated with the use of the IM classical mathematical model. In particular, this applies to the tested IM characterized by the intense rotor skin effect. Consequently, the obtained

results confirm the correct operation of the novel rotor flux estimation scheme and its robustness for electromagnetic parameter variability resulting from the rotor deep-bar effect.

## 2. Mathematical Models of an Induction Motor

One of the fundamental problems associated with the use of the IM classical mathematical model with constant parameters in IM control algorithms is the variability of motor electromagnetic parameters which is conditioned by changes of motor winding temperature, ferromagnetic core saturation, as well as the rotor deep-bar effect [26]. Figure 1a presents the T-type equivalent circuit corresponding to the IM classical mathematical model expressed in the Laplace-domain ($p$-domain), in which the rotor resistance $R_2$ and leakage inductance $L_{\sigma2}$ are represented by parameters varying as a function of slip frequency $\omega_2$. The variability of rotor electromagnetic parameters resulting from the rotor deep-bar effect can be modeled in the rotor equivalent circuit by a two-terminal network with constant parameters [16–25]. The electromagnetic parameters of such an IM mathematical model can be determined based on the $p$-domain motor inductance:

$$L_1(p) = \frac{\underline{\Psi}_{1r}(p)}{\underline{I}_{1r}(p)} = \omega_b \frac{Z_{ab}(p)}{p} \tag{1}$$

where $\underline{\Psi}_{1r}(p)$ and $\underline{I}_{1r}(p)$ are the Laplace transforms of the stator flux and current space vectors, respectively, $p$ is a complex frequency, $Z_{ab}(p)$ denotes the $p$-domain impedance between the terminals "a" and "b" of the IM equivalent circuit presented in Figure 1a, $\omega_b$ is the base frequency (Appendix A), and the subscript "r" denotes physical quantity space vectors expressed in an orthogonal coordinate system rotating at the shaft angular velocity $\omega_{sh}$.

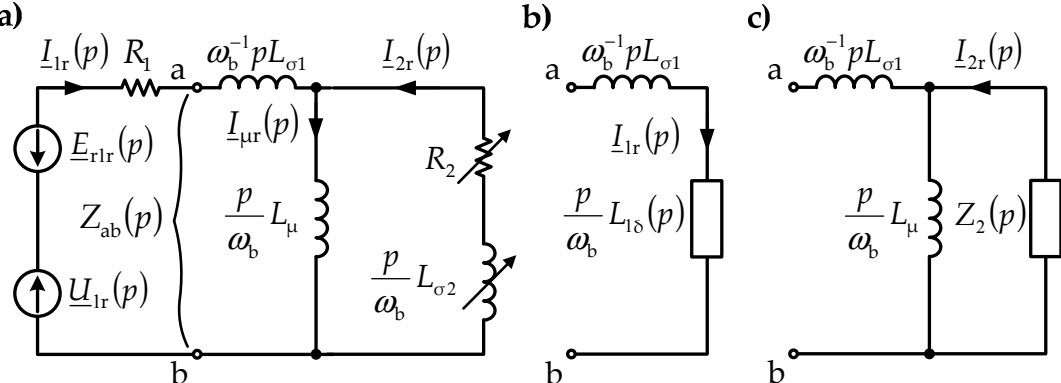

**Figure 1.** The induction motor (IM) equivalent circuits expressed in the Laplace-domain: (**a**) The IM equivalent circuit with rotor resistance and leakage inductance represented by parameters varying as a function of slip frequency. (**b**) The representation of the $p$-domain motor inductance by a series connection of the stator leakage inductance and the $p$-domain inductance associated with the airgap flux. (**c**) The representation of the $p$-domain inductance associated with the airgap flux by a parallel connection of the magnetizing inductance and the $p$-domain rotor impedance.

Equation (1), as well as the subsequent equations included in this paper, are expressed in the per-unit (p.u.) system. The base values of the used p.u. system are defined in Appendix A. Moreover, rotor physical quantities and electromagnetic parameters are referred to the stator.

The $p$-domain motor inductance $L_1(p)$ is a series connection of the stator leakage inductance $L_{\sigma1}$ and the $p$-domain inductance associated with the airgap flux $L_{1\delta}(p)$ (Figure 1b):

$$L_1(p) = L_{\sigma1} + L_{1\delta}(p). \tag{2}$$

The $p$-domain motor inductance $L_{1\delta}(p)$ can be further represented as a parallel connection of the magnetizing inductance $L_\mu$ and the $p$-domain rotor impedance $Z_2(p)$ (Figure 1c):

$$\frac{\omega_{\mathrm{b}}}{pL_{1\delta}(p)} = \frac{\omega_{\mathrm{b}}}{pL_{\mu}} + \frac{1}{Z_2(p)}. \tag{3}$$

The $p$-domain inductance $L_{1\delta}(p)$ can be derived from a solution of Maxwell's differential system of equations which are formulated, for instance, on the basis of an IM multi-layer model [17]. However, the $p$-domain inductance $L_{1\delta}(p)$ is not directly applicable in an analysis of IM transients due to the lack of possibility for inverse transformation of a Laplace transform including this inductance. The above-mentioned difficulty can be circumvent by the partial fraction decomposition of the inverse $p$-domain inductance $L_{1\delta}(p)$, which is an irrational function with an infinite number of negative real poles. This, in turn, leads to the rotor mathematical model in the form of a two-terminal network with constant $R_{2(n)}$, $L_{\sigma 2(n)}$ parameters [17]:

$$\frac{\omega_{\mathrm{b}}}{pL_{1\delta}(p)} = \frac{\omega_{\mathrm{b}}}{pL_{\mu}} + \sum_{n=1}^{\infty} \frac{1}{R_{2(n)} + \frac{1}{\omega_{\mathrm{b}}}pL_{\sigma 2(n)}}. \tag{4}$$

An exact approximation of the reference $p$-domain inductance $L_{1\delta}(p)$ is obtained with an infinite number of poles of an approximative rational function (an infinite number of parallel connected two-terminals in the rotor mathematical model). For the sake of the desired simplicity of the IM mathematical model, the number of parallel connected two-terminals in the rotor equivalent circuit is limited to $N$ two-terminals, and for achieving the required approximation accuracy of the irrational function $L_{1\delta}(p)$, the $(N + 1)$th residual two-terminal with parameters $R_{2(0)}$, $L_{\sigma 2(0)}$ is included (Figure 2) [17]:

$$\frac{\omega_{\mathrm{b}}}{pL_{1\delta}(p)} = \frac{\omega_{\mathrm{b}}}{pL_{\mu}} + \sum_{n=1}^{N} \frac{1}{R_{2(n)} + \frac{1}{\omega_{\mathrm{b}}}pL_{\sigma 2(n)}} + \frac{1}{R_{2(0)} + \frac{1}{\omega_{\mathrm{b}}}pL_{\sigma 2(0)}}. \tag{5}$$

The methodology for determination of the residual two-terminal electromagnetic parameters $R_{2(0)}$, $L_{\sigma 2(0)}$ has been described previously [17]. The approximation accuracy of the reference $p$-domain inductance is evaluated by comparing its frequency characteristic with a characteristic $\underline{L}_{1\delta}(p = \mathrm{j}\omega_2)$ resulting from the IM mathematical model with rotor two-terminal network representation.

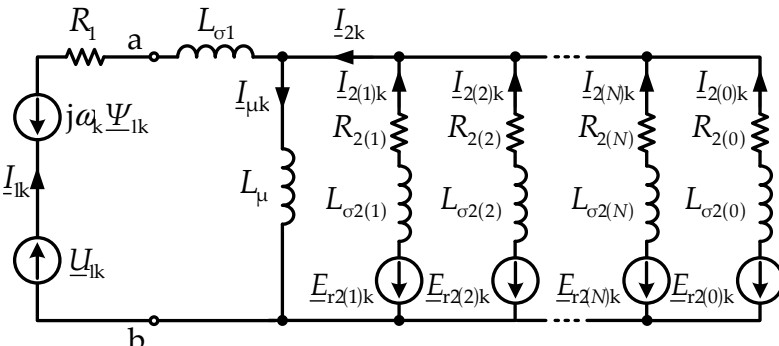

**Figure 2.** The equivalent circuit of an induction motor with rotor two-terminal network representation.

If the analytical formula describing the $p$-domain inductance $L_{1\delta}(p)$ is not known, which is the case, for instance, in experimental determination of the reference inductance frequency characteristic (IFCh) $\underline{L}_{1\delta}(\omega_2)$ [25,27], then the "synthetic" electromagnetic parameters of the IM equivalent circuit (Figure 2) can be identified as a result of an approximation of the reference IFCh $\underline{L}_{1\delta}(\omega_2)$ by the characteristic $\underline{L}_{1\delta}(p = \mathrm{j}\omega_2)$ derived from the adopted IM mathematical model. In such an approach, the residual two-terminal does not formally occur in Equation (5), and its participation in the approximation accuracy of the $p$-domain rotor impedance is smaller for larger the numbers $N$ of parallel connected two-terminals in the rotor mathematical model:

$$\frac{1}{Z_2(p)} \cong \sum_{n=1}^{N} \frac{1}{R_{2(n)} + \frac{1}{\omega_b} p L_{\sigma 2(n)}}. \tag{6}$$

In this way, an IM of any rotor construction (e.g., squirrel-cage, double-cage, or solid rotors) can be represented with the use of the mathematical model in which the electromagnetic parameter variability resulting from the rotor deep-bar effect is approximated by the two-terminal network with constant parameters. The process of electromagnetic parameter identification for individual two-terminals in the rotor equivalent circuit, ensuring the required approximation accuracy of the reference IFCh $\underline{L}_{1\delta}(\omega_2)$, can be conveniently performed using selected optimization methods. In such an approach, the number $N$ of the parallel connected two-terminals is determined empirically.

The IM mathematical model with rotor $N$-loop representation is described by the following system of equations:

$$\underline{U}_{1k} = R_1 \underline{I}_{1k} + T_N \frac{d}{dt} \underline{\Psi}_{1k} + j\omega_k \underline{\Psi}_{1k} \tag{7a}$$

$$\begin{cases} 0 = R_{2(1)} \underline{I}_{2(1)k} + T_N \frac{d}{dt} \underline{\Psi}_{2(1)k} + j(\omega_k - \omega_{sh}) \underline{\Psi}_{2(1)k} \\ \cdots \\ 0 = R_{2(N)} \underline{I}_{2(N)k} + T_N \frac{d}{dt} \underline{\Psi}_{2(N)k} + j(\omega_k - \omega_{sh}) \underline{\Psi}_{2(N)k} \end{cases} \tag{7b}$$

$$\underline{\Psi}_{1k} = L_1 \underline{I}_{1k} + L_\mu \underline{I}_{2k} \tag{7c}$$

$$\begin{cases} \underline{\Psi}_{2(1)k} = L_\mu \left( \underline{I}_{1k} + \underline{I}_{2k} \right) + L_{\sigma 2(1)} \underline{I}_{2(1)k} \\ \cdots \\ \underline{\Psi}_{2(N)k} = L_\mu \left( \underline{I}_{1k} + \underline{I}_{2k} \right) + L_{\sigma 2(N)} \underline{I}_{2(N)k} \end{cases} \tag{7d}$$

$$\underline{I}_{2k} = \sum_{n=1}^{N} \underline{I}_{2(n)k} \tag{7e}$$

$$\frac{d\omega_{sh}}{dt} = \frac{1}{T_M} (T_{em} - T_L) \tag{7f}$$

$$T_{em} = Im \left( \underline{\Psi}_{1k}^* \underline{I}_{1k} \right) \tag{7g}$$

$$\underline{E}_{r1k} = j\omega_k \underline{\Psi}_{1k} \tag{7h}$$

$$\underline{E}_{r2(n)k} = j(\omega_k - \omega_{sh}) \underline{\Psi}_{2(n)k} \tag{7i}$$

where $\underline{U}_{1k}$ and $\underline{E}_{r1k}$ are the stator voltage and electromotive force space vectors, respectively, $\underline{I}_{2(n)k}$, $\underline{\Psi}_{2(n)k}$, and $\underline{E}_{r2(n)k}$ represent the rotor current, flux, and electromotive force, respectively, related to the $n$th two-terminal in the rotor equivalent circuit, $T_{em}$ and $T_L$ constitute the electromagnetic and load torque, respectively, $T_N = 1/\omega_b$, $T_M$ is the motor mechanical time constant, $j^2 = -1$, * denotes the complex conjugate, and $k$ indicates physical quantity space vectors expressed in an orthogonal coordinate system rotating at an arbitrary angular velocity $\omega_k$.

## 3. The Novel Estimation Scheme of the Rotor Flux Space Vector

The variability of rotor electromagnetic parameters resulting from the rotor deep-bar effect can be represented in the rotor mathematical model by the two-terminal network with constant $R_{2(n)}$, $L_{\sigma 2(n)}$ parameters. Therefore, the application of such an IM mathematical model in an estimation scheme of the rotor flux space vector is justifiable, especially in the case of an IM characterized by the intense rotor deep-bar effect. However, the rotor flux estimation scheme, which would be based on the IM mathematical model of this type, has not been developed until now. This chapter presents the investigations leading to define the unique rotor flux space vector on the basis of the IM mathematical model with rotor two-terminal network representation.

The current space vector of the $n$th two-terminal in the rotor equivalent circuit (Figure 2) can be determined with the use of the transformed Equation (7d):

$$\underline{I}^{\bullet}_{2(n)\text{k}} = \frac{1}{L^{\bullet}_{\sigma2(n)}}\left(\underline{\Psi}^{\bullet}_{2(n)\text{k}} - L_\mu\underline{I}_{\mu\text{k}}\right) \tag{8a}$$

$$\underline{I}_{\mu\text{k}} = \underline{I}_{1\text{k}} + \underline{I}_{2\text{k}} \tag{8b}$$

$$\underline{\Psi}_{\mu\text{k}} = L_\mu\underline{I}_{\mu\text{k}} \tag{8c}$$

where $\underline{I}_{\mu\text{k}}$ and $\underline{\Psi}_{\mu\text{k}}$ are the magnetizing current and flux space vectors, respectively.

Incorporation of the formulas describing the rotor two-terminal current space vectors (Equation (8a)) to the transformed voltage equation (Equation (7b)) associated with the $n$th two-terminal in the rotor multi-loop equivalent circuit, the model of the rotor flux space vector related to the $n$th two-terminal is obtained in the form of:

$$T_{2(n)}\frac{\text{d}}{\text{d}t}\underline{\Psi}_{2(n)\text{k}} = L_\mu\underline{I}_{\mu\text{k}} - \underline{\Psi}_{2(n)\text{k}} + \text{j}\omega_\text{b}T_{2(n)}(\omega_\text{k} - \omega_\text{m})\underline{\Psi}_{2(n)\text{k}} \tag{9a}$$

$$T_{2(n)} = \frac{L_{\sigma2(n)}}{R_{2(n)}}T_\text{N} \tag{9b}$$

where $T_{2(n)}$ constitutes the time constant of the $n$th two-terminal in the rotor equivalent circuit presented in Figure 2.

The magnetizing current space vector $\underline{I}_{\mu\text{k}}$, which is required in Equation (9a), can be determined based on Equation (7c) where the stator flux space vector $\underline{\Psi}_{1\text{k}}$ is obtained with the use of the stator voltage Equation (7a):

$$\underline{I}_{\mu\text{k}} = \frac{1}{L_\mu}\left(\underline{\Psi}_{1\text{k}} - L_{\sigma1}\underline{I}_{1\text{k}}\right). \tag{10}$$

In general, the rotor flux space vector can be expressed as the sum of the magnetizing flux (Equation (8c)) and rotor leakage flux space vectors. Concerning the IM mathematical model with rotor multi-loop representation, the equation takes the following form:

$$\underline{\Psi}_{2\text{k}} = L_\mu\underline{I}_{1\text{k}} + \left(L_\mu + L_{\sigma2\text{eq}}\right)\underline{I}_{2\text{k}} \tag{11}$$

where $L_{\sigma2\text{eq}}$ is the equivalent rotor leakage inductance of the rotor two-terminal network (Figure 2).

The resultant rotor current space vector is the sum of the current space vectors of parallel connected two-terminals in the rotor equivalent circuit (Equation (7e)). Taking into account the formulas describing these current space vectors (Equation (8a)), Equation (11) is as follows:

$$\underline{\Psi}_{2\text{k}} = L_\mu\underline{I}_{\mu\text{k}} + L_{\sigma2\text{eq}}\sum_{n=1}^{N}\frac{\underline{\Psi}_{2(n)\text{k}}}{L_{\sigma2(n)}} - L_\mu\underline{I}_{\mu\text{k}}L_{\sigma2\text{eq}}\sum_{n=1}^{N}\frac{1}{L_{\sigma2(n)}}. \tag{12}$$

The magnetizing flux space vector $\underline{\Psi}_{\mu\text{k}}$ has been included in the formulas representing the flux space vectors associated with the individual two-terminals in the rotor multi-loop equivalent circuit (Equation (9a)), thus the magnetizing flux space vector is redundant in Equation (12) for the rotor flux space vector. The first and the third components of the sum in Equation (12), constituting and containing the magnetizing flux space vector, respectively, reduce each other in cases when the inverse equivalent rotor leakage inductance $L_{\sigma2\text{eq}}$ equals the sum of the inverse leakage inductances of the individual rotor two-terminals:

$$\frac{1}{L_{\sigma2\text{eq}}} = \sum_{n=1}^{N}\frac{1}{L_{\sigma2(n)}}. \tag{13}$$

On the basis of the above reasoning, the derived formulas describe the equivalent rotor leakage inductance of the rotor equivalent circuit presented in Figure 2, which result from a parallel connection of leakage inductances of the individual rotor two-terminals:

$$L_{\sigma 2 \text{eq}} = \lim_{p \to \infty} \frac{Z_2(p)}{p} \tag{14}$$

and the unique rotor flux space vector of the IM mathematical model with rotor multi-loop representation:

$$\underline{\Psi}_{2\text{k}} = L_{\sigma 2 \text{eq}} \sum_{n=1}^{N} \frac{\underline{\Psi}_{2(n)\text{k}}}{L_{\sigma 2(n)}}. \tag{15}$$

According to the above, the voltage–current model of the rotor flux space vector can be formulated. When expressed in the orthogonal coordinate system ($\alpha$–$\beta$) stationary with respect to the stator ($\omega_\text{k} = 0$, indicated by *s*), this model is represented by the following system of equations:

$$\underline{I}_{\mu\text{s}}^{\text{e}} = \frac{1}{L_\mu}\left( \frac{1}{T_\text{N}} \int_0^t \left(\underline{U}_{1\text{s}} - R_1 \underline{I}_{1\text{s}}\right) \mathrm{d}t - L_{\sigma 1} \underline{I}_{1\text{s}} \right) \tag{16a}$$

$$\begin{cases} T_{2(1)} \frac{\mathrm{d}}{\mathrm{d}t} \underline{\Psi}_{2(1)\text{s}}^{\text{e}} = L_\mu \underline{I}_{\mu\text{s}}^{\text{e}} - \underline{\Psi}_{2(1)\text{s}}^{\text{e}} + \mathrm{j}\omega_\text{b} T_{2(1)} \omega_\text{m} \underline{\Psi}_{2(1)\text{s}}^{\text{e}} \\ \cdots \\ T_{2(N)} \frac{\mathrm{d}}{\mathrm{d}t} \underline{\Psi}_{2(N)\text{s}}^{\text{e}} = L_\mu \underline{I}_{\mu\text{s}}^{\text{e}} - \underline{\Psi}_{2(N)\text{s}}^{\text{e}} + \mathrm{j}\omega_\text{b} T_{2(N)} \omega_\text{m} \underline{\Psi}_{2(N)\text{s}}^{\text{e}} \end{cases} \tag{16b}$$

$$\underline{\Psi}_{2\text{s}}^{\text{e}} = L_{\sigma 2 \text{eq}} \sum_{n=1}^{N} \frac{\underline{\Psi}_{2(n)\text{s}}^{\text{e}}}{L_{\sigma 2(n)}} \tag{16c}$$

where "e" denotes the estimated rotor flux space vector.

Figure 3 presents a schematic diagram of the rotor flux estimation scheme, corresponding to Equations (16a)–(16c). The rotor angular velocity and the stator voltage and current space vector components constitute the input signals of the developed rotor flux estimation scheme. It should also be noted that when the rotor deep-bar effect is represented by the single two-terminal $N = 1$ in the rotor mathematical model, the voltage-current model of the rotor flux space vector remains valid and corresponds, in this case, to the IM classical mathematical model.

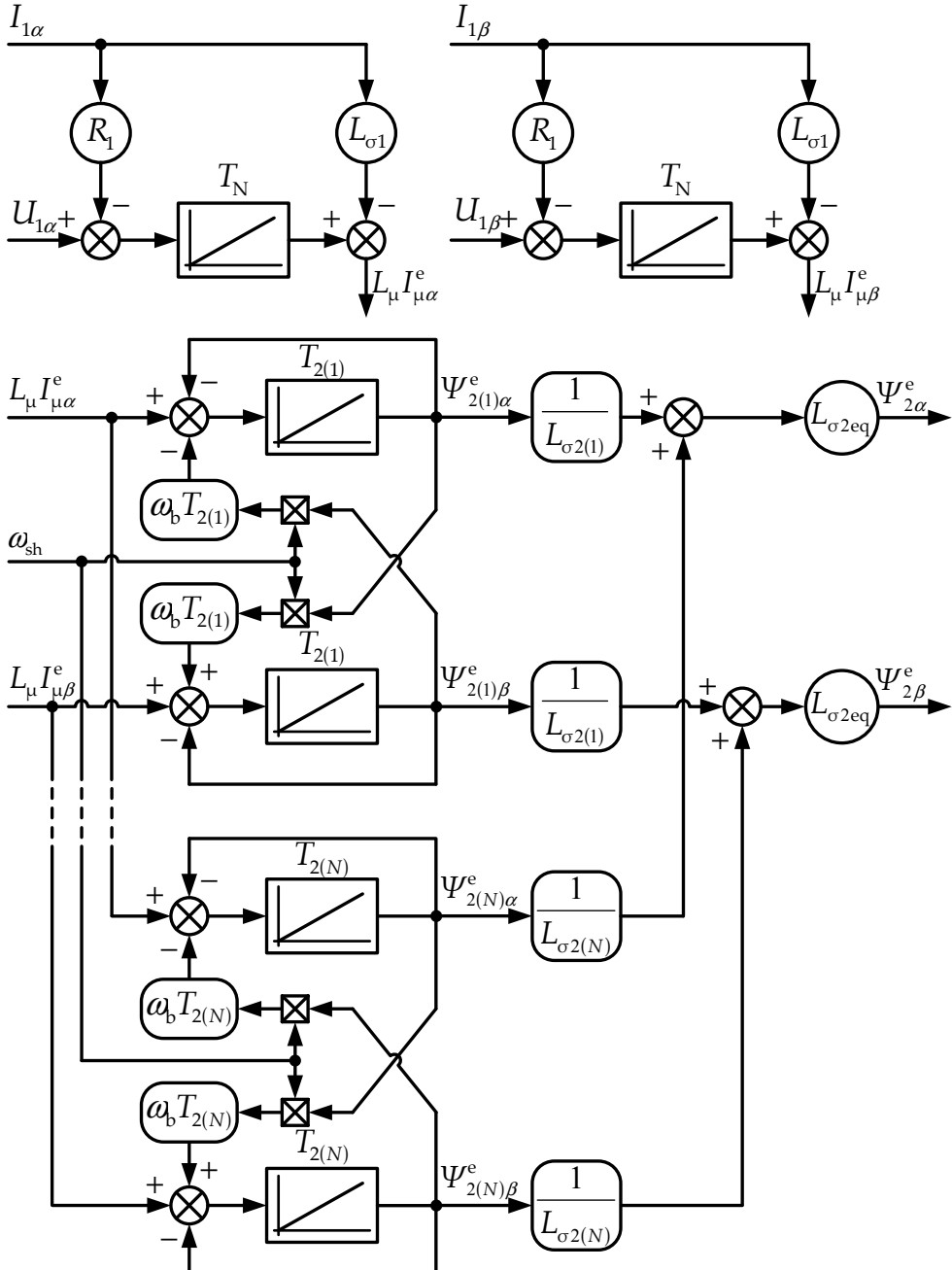

**Figure 3.** The schematic diagram of the voltage–current model of the rotor flux space vector, developed on the basis of the IM mathematical model with rotor two-terminal network representation, expressed in the orthogonal coordinate system ($\alpha$–$\beta$) stationary with respect to the stator $\omega_k = 0$.

## 4. Laboratory Tests

### 4.1. Laboratory Test Bench

The verification of the novel rotor flux estimation scheme was performed with the laboratory test bench, of which a schematic diagram is shown in Figure 4. Figure 5 presents a picture of the laboratory test bench.

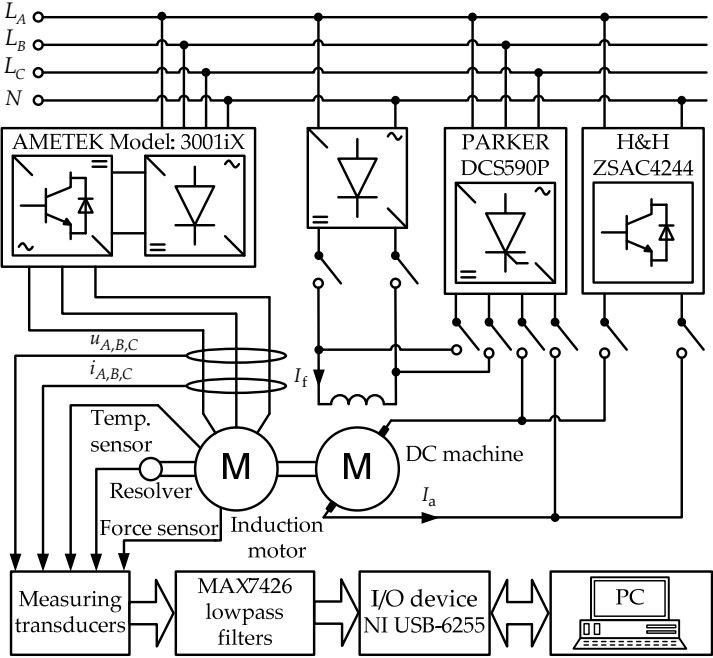

**Figure 4.** The schematic diagram of the laboratory test bench.

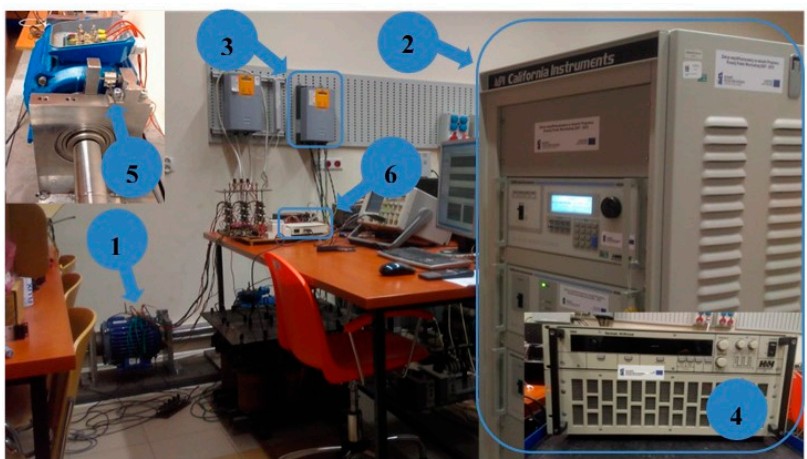

**Figure 5.** The laboratory test bench. The particular markers indicate: 1. tested induction motor, 2. programmable AC power source, AMETEK model: 3001iX, 3. 4Q thyristor converter, Parker DC590P, 4. electronic AC load ZSAC4244, Höcherl and Hackl GmbH, 5. tensometric force sensor, 6. multifunction I/O device NI USB-6255.

Investigations were conducted on the four-pole IMs of type Sg 132S-4 with a squirrel-cage rotor (CR-IM) and a solid rotor (SR-IM). The cross-section dimensions of the studied rotors are included in Appendix B.

The solid rotor was designed and manufactured only for the purpose of the presented study. This is because of the fact that such a rotor is characterized by the intense skin effect. This feature was conveniently used in the verification of the novel rotor flux estimation scheme. Such an approach allowed us to conduct experimental investigations with the use of low-power IMs, giving the background for the employment of the large squirrel-cage IMs in the next stage of the study. The tested SR-IM is marked by significant slip $s \approx 1$ corresponding to the breakdown torque at the stator supply voltage frequency of $f_1 = 50$ Hz. In order to reduce the breakdown slip of the SR-IM, the new operating points were adopted for both the CR-IM and SR-IM. The rated values of the tested IMs corresponding to the adopted machine operating points were set together in Appendix B.

The tested IMs (marker 1 in Figure 5) were powered by the programmable AC source AMETEK Model: 3001iX (marker 2 in Figure 5). The investigations were carried out at the three setpoints of stator voltage frequency for both considered IMs, maintaining a constant voltage/frequency ratio. Prior to the measurement tests, the CR-IM and SR-IM operated at the given load conditions for a period of time, allowing the stator winding temperature to stabilize at the assumed level. This aimed to reduce the influence of variability of rotor and stator resistances, resulting from winding temperature changes, on the identified electromagnetic parameters of the IM mathematical models and on the accuracy evaluation of the rotor flux space vector estimation.

A separately excited DC machine of type PCMb 54b served as a load for the investigated IMs in the presented study. During the no-load, blocked rotor, and load curve (LC) tests, conducted in order to identify the electromagnetic parameters of the considered IM mathematical models, the DC machine was powered by the 4Q thyristor converter Parker DC590P (marker 3 in Figure 5). Such a solution provided a wide adjustment range of load conditions for the tested IMs, enabling the measurement of demanded physical quantities in the generating, motoring, and ideal no-load modes of machine operation. In turn, the programmable electronic load ZSAC4244 – H&H GmbH (marker 4 in Figure 5) was used to control the armature current of the separately excited DC machine for the verification of the rotor flux estimation scheme. Such an approach enabled shaping of the desired dynamics of IM slip frequency changes in the assumed range, corresponding to the load adjustment range up to 1.30 of the stator rated current for both tested IMs. The slip frequency range considered during the verification of the rotor flux estimation scheme corresponded also to the slip frequency range when the LC tests were conducted. The power rating of the individual devices and DC machine which were used in the experimental investigations are included in Appendix B.

During the laboratory tests, the measurement of stator winding voltages, currents, and temperature, as well as shaft angular velocity, were carried out. Additionally, the shaft torque of the investigated IMs was determined on the basis of the force measurement realized by means of the force sensor (marker 5 in Figure 5). The accuracy class and measuring range of the used force sensor were 0.2 and 5 kN, respectively. The stator currents of the tested IMs were converted into voltage signals by means of non-inductive resistive voltage dividers with an accuracy class of 0.5 and a measuring range of 10 A. Similarly, the stator voltages were scaled by means of voltage dividers with a voltage ratio of 1000:1, composed of non-inductive resistors with an accuracy class of 0.2. The angular velocity measurement was carried with the use of a resolver. Data acquisition (DAQ) was performed by means of the National Instruments USB-6255 high-resolution, multifunction I/O device (marker 6 in Figure 5). The DAQ system was equipped with the MAX7426 5th-order, lowpass, elliptic, switched-capacitor filters. The configuration of the DAQ device and the acquisition of measurement data were carried out in the National Instruments LabView environment.

*4.2. The Identification Procedure of Electormagnetic Parametersfor the IM Mathematical Model*

The identification process for electromagnetic parameters of the IM mathematical model with rotor two-terminal network representation was conducted in conformity to a procedure described previously [25].The reference IFCh $\underline{L}_1(\omega_2)$ of the considered IMs were determined on the basis of the measurement data derived from the LC test [28] according to the following equations:

$$\frac{\omega_\mathrm{b}}{\mathrm{j}\omega_1}\left(\underline{Z}_1(\omega_2) - R_1\right) = \underline{L}_1(\omega_2) = L_{1\sigma} + \underline{L}_{1\delta}(\omega_2) \tag{17a}$$

$$\underline{Z}_1(\omega_2) = \frac{\left|\underline{U}_{1\mathrm{ph}}\right|}{\left|\underline{I}_{1\mathrm{ph}}(\omega_2)\right|}\left(\cos(\phi_1(\omega_2)) + \mathrm{j}\sqrt{1 - [\cos(\phi_1(\omega_2))]^2}\right) \tag{17b}$$

$$\cos(\phi_1(\omega_2)) = \frac{P_1(\omega_2)}{3\left|\underline{U}_{1\mathrm{ph}}\right|\left|\underline{I}_{1\mathrm{ph}}(\omega_2)\right|} \tag{17c}$$

where $\underline{Z}_1(\omega_2)$ is the IM impedance expressed as a function of slip frequency, $\omega_1$ represents the stator voltage angular frequency, $|\underline{U}_{1\text{ph}}|$ and $|\underline{I}_{1\text{ph}}(\omega_2)|$ constitute the measured root mean squared values of stator phase voltages and currents, respectively, and $P_1(\omega_2)$ and $\cos(\phi_1(\omega_2))$ are the measured stator power and calculated stator power factor, respectively.

Approximation of the reference IFCh $\underline{L}_{1\delta}(\omega_2)$ by means of the frequency-domain inductance $\underline{L}_{1\delta}(p=\mathrm{j}\omega_2)$ resulting from the adopted IM mathematical model, for instance, with $N$ parallel connected two-terminals in the rotor equivalent circuit, in the form of the following equations:

$$\frac{1}{\underline{L}_{1\delta}(\omega_2)} \cong \frac{1}{\underline{L}_{1\delta}(p=\mathrm{j}\omega_2)} = \frac{1}{L_\mu} + \frac{\mathrm{j}\omega_2}{\omega_\mathrm{b}}\frac{1}{\underline{Z}_2(p=\mathrm{j}\omega_2)} \tag{18a}$$

$$\frac{1}{\underline{Z}_2(p=\mathrm{j}\omega_2)} \cong \sum_{n=1}^{N} \frac{1}{R_{2(n)} + \frac{\mathrm{j}\omega_2}{\omega_\mathrm{b}}L_{\sigma 2(n)}}. \tag{18b}$$

allows determination of the "synthetic" electromagnetic parameters of the IM mathematical model.

The stator phase winding resistance $R_1$ is identified through the DC line-to-line resistance measurement conducted according to the standards [28,29]. The parameters $L_{1\sigma}$, $L_\mu$, $R_{2(n)}$, and $L_{\sigma 2(n)}$ are subject to the identification process which can be considered as a minimization issue of the evaluation function, defined as the sum of the mean squared errors of the reference characteristic approximation [25]:

$$F\big(|\underline{L}_1(\omega_2)|, \angle\underline{L}_1(\omega_2)\big) = \sum_{\omega_{2\min}}^{\omega_{2\max}} k_{\mathrm{mod}}\left(\frac{|\underline{L}_1(\omega_2)| - |\underline{L}_1(p=\mathrm{j}\omega_2)|}{|\underline{L}_1(\omega_2)|}\right)^2 +$$
$$+ \sum_{\omega_{2\min}}^{\omega_{2\max}} k_{\mathrm{arg}}\big(\angle\underline{L}_1(\omega_2) - \angle\underline{L}_1(p=\mathrm{j}\omega_2)\big)^2 \tag{19}$$

where $|\underline{L}_1(\omega_2)|$ and $\angle\underline{L}_1(\omega_2)$ are the modulus and argument of the IM reference IFCh, respectively, $|\underline{L}_1(p=\mathrm{j}\omega_2)|$ and $\angle\underline{L}_1(p=\mathrm{j}\omega_2)$ constitute the modulus and argument of the frequency-domain IM inductance, respectively, $\omega_{2\min}$ and $\omega_{2\max}$ represent the lower and upper limits of the considered slip frequency range, and $k_{\mathrm{mod}}$ and $k_{\mathrm{arg}}$ are the weighting factors of the individual components of the evaluation function.

Similarly to the studies presented previously [25], a minimization process of the adopted evaluation function was carried out with the use of the genetic algorithm by means of the Genetic Algorithms for Optimization Toolbox in the Matlab environment. The choice of the genetic algorithm was dictated by the effectiveness of this optimization tool, as indicated in numerous scientific publications concerned the identification of electromagnetic parameters for IM mathematical models [30,31].

The criterion adopted in the identification process of electromagnetic parameters for the CR-IM and SR-IM mathematical models with rotor multi-loop representation, assumed the approximation of the reference IFCh modulus with an error not exceeding 2% in the considered range of slip frequency, while maintaining a possible minimum approximation error of the reference IFCh argument and a minimum number $N$ of two-terminals in a rotor equivalent circuit. The criterion was met with the use of the IM mathematical models with two $N=2$ and three $N=3$ parallel connected two-terminals in the cage and solid rotor network representations, respectively. The electromagnetic parameters of the IM mathematical models with multi-loop representation of the cage and solid rotors are denoted as CR-RML and SR-RML, respectively. These parameters are listed in Tables 1 and 2. Tables 1 and 2 also include the electromagnetic parameters of the IM classical mathematical model (T-type equivalent circuit parameters), which were identified based on selected procedures described in the standard 1 [28] (the designations are CR-T Std 1, SR-T Std 1) and standard 2 [29] (the designations are CR-T Std 2, SR-T Std 2) in the vicinity of the adopted operating points of the tested IMs (Appendix B). The stator phase resistance, determined according to the guidelines included in standard 1 [28], after correction to the reference winding temperature of 25 °C, equalled $R_1 = 2.9597\ \Omega$.

**Table 1.** Electromagnetic parameters (p.u.) of the considered squirrel-cage rotor induction motor (CR-IM) mathematical models. The individual resistances were corrected to the reference winding temperature of 25 °C.

| Electromagnetic Parameters | $L_{1\sigma}$ | $L_\mu$ | $R_{2(1)}$ | $L_{\sigma 2(1)}$ | $R_{2(2)}$ | $L_{\sigma 2(2)}$ |
|---|---|---|---|---|---|---|
| CR-T Std 1 | 0.0949 | 3.0978 | 0.0309 | 0.1428 | – | – |
| CR-T Std 2 | 0.0910 | 3.1235 | 0.0335 | 0.1358 | – | – |
| CR-RML | 0.1090 | 3.0203 | 0.0395 | 0.0888 | 0.1326 | 1.3294 |

**Table 2.** Electromagnetic parameters (p.u.) of the considered solid rotor induction motor (SR-IM) mathematical models. The individual resistances correspond to the average temperature of stator winding of 55 °C registered under the load curve(LC) test.

| Electromagnetic Parameters | $L_{1\sigma}$ | $L_\mu$ | $R_{2(1)}$ | $L_{\sigma 2(1)}$ | $R_{2(2)}$ | $L_{\sigma 2(2)}$ | $R_{2(3)}$ | $L_{\sigma 2(3)}$ |
|---|---|---|---|---|---|---|---|---|
| SR-T Std 1 | 0.2597 | 2.8323 | 0.1329 | 0.4550 | – | – | – | – |
| SR-T Std 2 | 0.1645 | 3.1736 | 0.2271 | 0.2456 | – | – | – | – |
| SR-RML | 0.1977 | 3.4401 | 0.2852 | 0.2313 | 0.4695 | 2.1032 | 0.1518 | 7.4735 |

As an example, in Figure 6, the SR-IM reference IFCh $\underline{L}_1(\omega_2)$ are set together with the approximative characteristics, which were determined on the basis of the IM mathematical model with single and three parallel connected two-terminals in the solid rotor equivalent circuit. Figure 7 presents the modulus relative errors and the argument absolute errors between the reference and approximative characteristics [25].

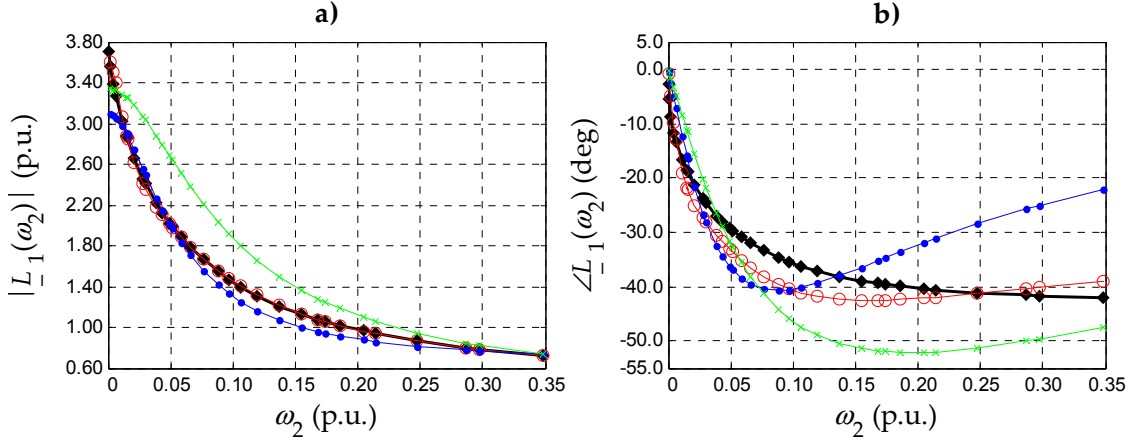

**Figure 6.** The SR-IM reference inductance frequency characteristic (IFCh) and its approximation by means of the IM mathematical models with single and three two-terminals in the solid rotor equivalent circuit: (**a**) Modulus and (**b**) argument.

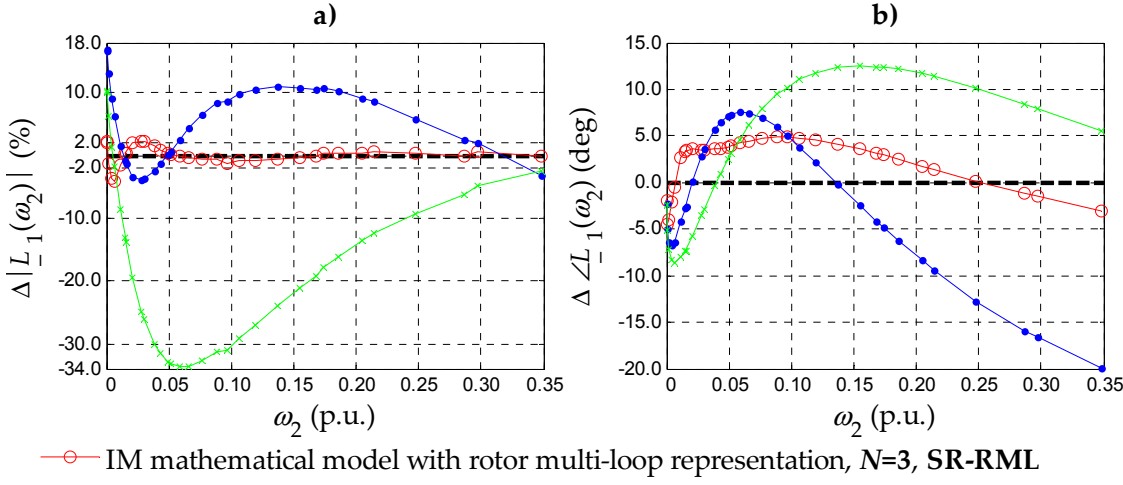

**Figure 7.** Approximation errors of the SR-IM reference IFCh: (**a**) Modulus relative errors and (**b**) argument absolute errors.

It is worth nothing that the use of the IM mathematical model with three parallel connected two-terminals in the solid rotor equivalent circuit allowed the most accurate approximation of the SR-IM reference IFCh to be achieved, in comparison to the approximation accuracy obtained by the IM classical mathematical model. The required approximation accuracy of the SR-IM reference IFCh modulus was met with argument approximation error not exceeding five degrees in the considered range of slip frequency (Figure 7b). Theoretically, approximation accuracy of the reference IFCh could be further improved by incorporating additional two-terminals in the rotor mathematical model, but the adopted criterion also assumed their minimal number. This resulted from the desirable simplicity of the rotor flux estimation scheme, which is intended for the IM rotor-field-oriented control implementation. Ultimately, the approximation of the SR-IM reference IFCh realized by the IM mathematical model with three parallel connected two-terminals in the solid rotor equivalent circuit was considered to be sufficiently accurate for the purpose of the rotor flux estimation.

### 4.3. Estimation of the Rotor Flux Space Vector

In the presented study, the estimation accuracy of the rotor flux space vector was evaluated in the slip frequency range corresponding to the IM load adjustment range up to 1.30 of the stator rated current. The range of slip frequency exceeding the typical operating range of slip frequency for the field-oriented controlled IM, was adopted in this study for the robustness verification of the novel rotor flux estimation scheme against electromagnetic parameter variability resulting from the rotor deep-bar effect. This is the reason why the developed rotor flux estimation scheme was not employed in the rotor-flux-oriented control at this stage of the study. Since the direct measurement of rotor flux is not realized in practice, the verification of the rotor flux estimation accuracy was conducted indirectly, based on the reference quantities registered with the laboratory test bench.

During the study, the every endeavor was made to conduct the measurements of the reference quantities in a manner to assure the minimal possible measurement uncertainty. Due to the high resolution of the DAQ device and the precision of the sensors and measuring transducers, the registered shaft torque $T_{sh}$ and the determined power losses $\Delta P$ were considered as the reference quantities in the verification of the developed rotor flux estimation scheme. The verification investigations used the fact that the shaft torque can be determined as the quotient of the motor shaft power and angular velocity:

$$T_{sh}^{e} = \frac{P_{sh}^{e}}{\omega_{sh}} \qquad (20a)$$

$$P_{\mathrm{sh}}^{\mathrm{e}} = T_{\mathrm{em}}^{\mathrm{e}}\omega_{\mathrm{sh}} - \Delta P \tag{20b}$$

where $T_{\mathrm{sh}}^{\mathrm{e}}$ and $T_{\mathrm{me}}^{\mathrm{e}}$ represent the estimated shaft and electromagnetic torque, respectively and $P_{\mathrm{sh}}^{\mathrm{e}}$ is the estimated shaft power.

The power losses $\Delta P$ occurring in Equation (20b) were determined according to Equations (21a)–(21c) based on the measurement data derived from the LC tests. In order to eliminate the necessity to split up individual components of the power losses $\Delta P$ at the considered load conditions of the tested IMs, core losses were not erased from the IM input power whilst calculating the airgap power $P_\delta$ (Equation (21c)). Such an approach is in line with the adopted IM mathematical model described by Equations (7a)–(7i), in which the resistance associated with core losses is not included. In the presented study, the power losses $\Delta P$ were considered as any power losses determining the difference between the power transferred to the shaft $(1 - s)P_\delta$ and the shaft power $P_{\mathrm{sh}}$:

$$\Delta P(s) = (1 - s)P_\delta(s) - P_{\mathrm{sh}}(s) \tag{21a}$$

$$P_{\mathrm{sh}}(s) = \frac{\omega_1}{\omega_{\mathrm{b}}}(1 - s)T_{\mathrm{sh}}(s) \tag{21b}$$

$$P_\delta(s) = \frac{3}{2}\mathrm{Re}\left(\underline{U}_1\underline{I}_1^*(s)\right) - \frac{3}{2}R_1\left|\underline{I}_1(s)\right|^2 \tag{21c}$$

where $P_\delta(s)$ constitutes the airgap power and $s$ represent the motor slip.

Figure 8 presents the variability of the power losses, the shaft power, and the power transferred to the shaft expressed as a function of angular velocity of the tested CR-IM (Figure 8a) and SR-IM (Figure 8b), at the stator voltage frequency of $\omega_1 = 1.0$ (p.u.). Due to slight changes of the CR-IM power losses in the considered slip frequency range (Figure 8a), a constant value of these losses $\Delta P = 0.0863$ (p.u.) was assumed in the verification studies of the novel rotor flux estimation scheme. In relation to the SR-IM, on account of significant changes of the power losses in the considered slip frequency range (Figure 8b), the power loss variability was approximated by means of the second order polynomial:

$$\Delta P(\omega_{\mathrm{sh}}) = -0.6551\omega_{\mathrm{sh}}^2 + 0.8076\omega_{\mathrm{sh}} - 0.0232. \tag{22}$$

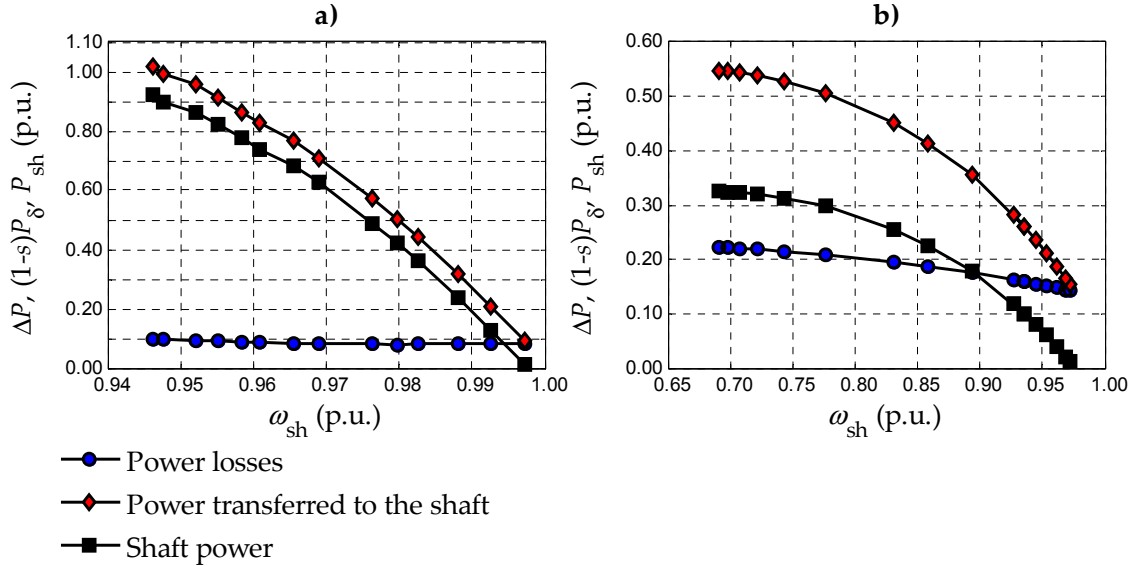

**Figure 8.** The variability of the power losses, the power transferred to the shaft and the shaft power expressed as a function of angular velocity of the investigated: (**a**) CR-IM and (**b**) SR-IM.

The electromagnetic torque required in Equation (20b) was determined with the use of the estimated rotor flux space vector in conformity with the following equations:

$$T_{\text{em}}^{\text{e}} = \frac{L_\mu}{L_{2\text{eq}}} \text{Im}\left( \underline{I}_{1\text{k}} \left( L_{\sigma 2\text{eq}} \sum_{n=1}^{N} \frac{\underline{\Psi}_{2(n)\text{k}}}{L_{\sigma 2(n)}} \right)^{*} \right) \tag{23a}$$

$$L_{2\text{eq}} = L_\mu + L_{\sigma 2\text{eq}}. \tag{23b}$$

The estimation precision of the registered shaft torque was verified based on the absolute estimation errors:

$$\Delta T_{\text{sh}}^{\text{e}} = T_{\text{sh}} - T_{\text{sh}}^{\text{e}}. \tag{24}$$

Additionally, Tables 3 and 4 present the maximum and mean absolute errors of the registered shaft torque estimation which were determined in accordance with the following equations:

$$\max\left|\Delta T_{\text{sh}}^{\text{e}}\right| = \max\left|T_{\text{sh}} - T_{\text{sh}}^{\text{e}}\right| \tag{25a}$$

$$\text{M}\left|\Delta T_{\text{sh}}^{\text{e}}\right| = \frac{1}{n} \sum_{i=1}^{n} \left|T_{\text{sh},i} - T_{\text{sh},i}^{\text{e}}\right| \tag{25b}$$

where $T_{\text{sh},i}$ and $T_{\text{sh},i}^{\text{e}}$ represent the *i*th samples of the registered and estimated shaft torque, respectively, and *n* is the number of samples.

For the sake of comparison, the shaft torque estimated through the use of the so called full order open-loop flux observer [32] was also considered in the presented verification. Research results presented previously [32] indicate that this rotor flux estimation scheme is characterized by limited sensitivity to erroneous identification or variability of the rotor electromagnetic time constant in a wide range of slip frequency changes, in comparison to the commonly known current model of the rotor flux space vector. The full order open-loop flux observer was formulated on the basis of the IM classical mathematical model, and is represented by the following system of equations:

$$T_{\text{N}} \frac{\text{d}}{\text{d}t} \underline{I}_{1\text{s}}^{\text{e}} = \frac{1}{\sigma L_1} \left[ \underline{U}_{1\text{s}} - \left( R_1 + \left(\frac{L_\mu}{L_2}\right)^2 R_2 \right) \underline{I}_{1\text{s}}^{\text{e}} + \frac{L_\mu}{L_2}\left(\frac{R_2}{L_2} - \text{j}\omega_{\text{m}}\right)\underline{\Psi}_{2\text{s}}^{\text{e}} \right] \tag{26a}$$

$$T_2 \frac{\text{d}}{\text{d}t} \underline{\Psi}_{2\text{s}}^{\text{e}} = L_\mu \underline{I}_{1\text{s}}^{\text{e}} - \underline{\Psi}_{2\text{s}}^{\text{e}} + \text{j}\omega_{\text{b}} T_2 \omega_{\text{m}} \underline{\Psi}_{2\text{s}}^{\text{e}} \tag{26b}$$

$$T_2 = \frac{L_2}{R_2} T_{\text{N}} \tag{26c}$$

where $T_2$ is the rotor electromagnetic time constant.

The verification of the rotor flux estimation schemes additionally includes the shaft torque, which was estimated with the help of the elaborated voltage–current model (Equations (16a)–(16c)) and formulated on the basis of the IM classical mathematical model.

Figure 9 presents a block diagram of the algorithm used in the accuracy evaluation of the rotor flux space vector estimation. The algorithm was implemented in the Matlab environment.

In the presented study, the investigated estimation schemes of the rotor flux space vector were fed by the registered angular velocity and stator voltages and currents. This case corresponded to the operation of the tested estimation schemes in an IM drive with angular velocity measurements. Due to the similar accuracy of the shaft torque estimation, obtained through the considered estimation schemes of the rotor flux space vector at each setpoint of stator voltage frequency for both tested IMs, this paper presents the study results conducted at the nominal stator voltage frequency $\omega_1 = 1.0$ (p.u.).

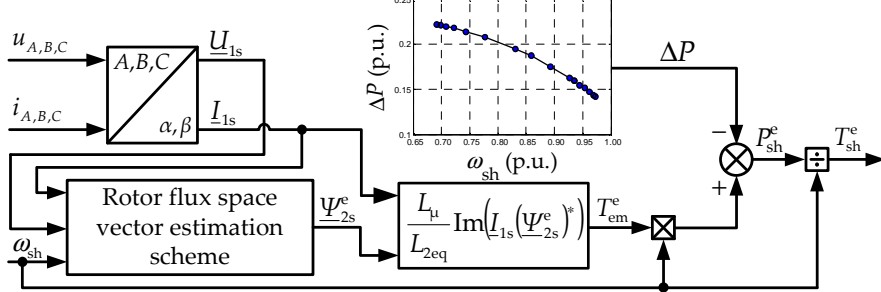

**Figure 9.** The block diagram of the algorithm used in the accuracy evaluation of the rotor flux space vector estimation.

Figure 10a presents the registered SR-IM shaft torque together with the shaft torque generated with the use of the tested rotor flux estimation schemes which were formulated on the basis of the IM mathematical model with single and three parallel connected two-terminals in the rotor equivalent circuit. The absolute errors of the registered shaft torque estimation are shown in Figure 10b, whereas Table 3 lists the maximal and mean absolute estimation errors. Figure 10 includes the shaft torque obtained with the use of the rotor flux estimation schemes based on the IM classical mathematical model in the structure of which the electromagnetic parameters SR-Std 2 were applied. The use of these parameters allowed for a more accurate estimation of the registered SR-IM shaft torque in relation to the estimation accuracy acquired by means of the employed schemes with the electromagnetic parameters SR-Std 1 (Table 3).

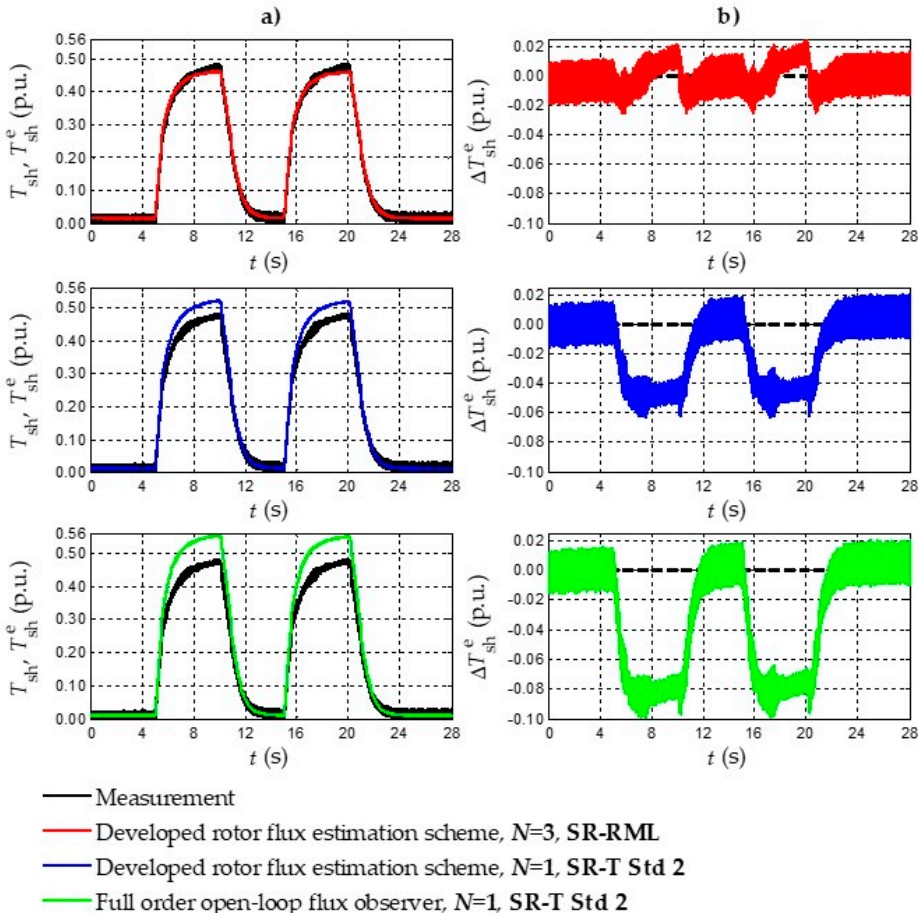

**Figure 10.** The SR-IM shaft torque estimation: (**a**) The registered and estimated shaft torque and (**b**) the absolute errors of the registered shaft torque estimation.

**Table 3.** Maximum and mean absolute errors (p.u.) of the registered shaft torque estimation of the tested SR-IM.

| Rotor Flux Estimation Schemes | Full Order Open-Loop Flux Observer (26a)–(26c) | | Voltage-Current Model (16a)–(16c) $N = 1$ | | Voltage-Current Model (16a)–(16c) $N = 3$ |
|---|---|---|---|---|---|
| **Electromagnetic parameters** | SR-T Std 1 | SR-T Std 2 | SR-T Std 1 | SR-T Std 2 | SR-RML |
| $\max|\Delta T_{sh}^e|$ | 0.3191 | 0.0986 | 0.2877 | 0.0631 | 0.0262 |
| $M|\Delta T_{sh}^e|$ | 0.0845 | 0.0346 | 0.0712 | 0.0209 | 0.0075 |

Figure 11a presents the registered and estimated shaft torque of the tested CR-IM, whereas Figure 11b shows the absolute estimation errors. The maximal and mean absolute estimation errors are listed in Table 4.

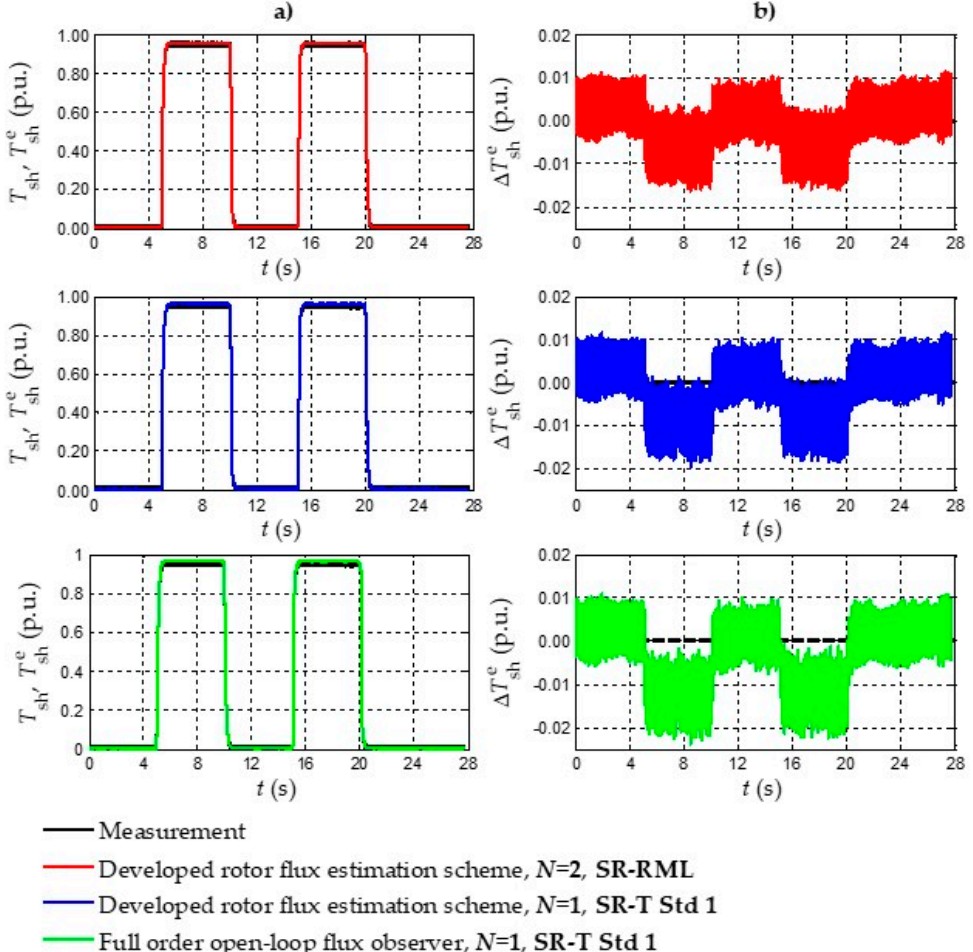

**Figure 11.** The CR-IM shaft torque estimation: (**a**) The registered and estimated shaft torque and (**b**) the absolute errors of the registered shaft torque estimation.

**Table 4.** Maximum and mean absolute errors (p.u.) of the registered shaft torque estimation of the tested CR-IM.

| Rotor Flux Estimation Schemes | Full Order Open-Loop Flux Observer (26a)–(26c) | | Voltage-Current Model (16a)–(16c) $N = 1$ | | Voltage-Current Model (16a)–(16c) $N = 2$ |
|---|---|---|---|---|---|
| **Electromagnetic parameters** | CR-T Std 1 | CR-T Std 2 | CR-T Std 1 | CR-T Std 2 | CR-RML |
| $\max|\Delta T_{sh}^e|$ | 0.0239 | 0.0345 | 0.0196 | 0.0211 | 0.0164 |
| $M|\Delta T_{sh}^e|$ | 0.0065 | 0.0105 | 0.0056 | 0.0063 | 0.0047 |

As regards the CR-IM, a more accurate estimation of the registered shaft torque was achieved using the electromagnetic parameters CR-T Std 1 in the considered estimation schemes based on the IM classical mathematical model, in comparison to when the electromagnetic parameters CR-T Std 2 were applied (Table 4). For this reason, the shaft torque generated through the rotor flux estimation schemes described by Equations (26a)–(26c) and (16a)–(16c) (single two-terminal rotor representation $N = 1$) with the electromagnetic parameters CR-T Std 1 are included in Figure 11.

## 5. Conclusions

This paper presents a novel estimation scheme of the rotor flux space vector which has been developed on the basis of the IM mathematical model with rotor multi-loop representation. In regards to the tested SR-IM, the use of the rotor flux estimation scheme, in which the rotor skin effect was modeled by three parallel connected two-terminals in the rotor equivalent circuit, enabled a multiple reduction of the registered shaft torque estimation errors in relation to the estimation errors obtained through the considered estimation schemes based on the IM classical mathematical model (Table 3). The absolute estimation errors of the registered SR-IM shaft torque achieved by using the elaborated voltage–current model ($N = 3$) did not exceed the level of ±0.0262 (p.u.) (Figure 10b, Table 3). The results of the presented study indicate considerable improvement in the accuracy of the rotor flux space vector estimation of the tested SR-IM, which was obtained by the estimation scheme elaborated on the IM mathematical model with rotor two-terminal network representation, in comparison with the estimation precision acquired by the schemes formulated on the IM classical mathematical model.

It should also be noted that even for the tested CR-IM, which does not show a substantial deep-bar effect, the superiority of the novel rotor flux estimation scheme (with two parallel connected two-terminals $N= 2$ in the rotor equivalent circuit) over the estimation schemes based on the IM classical mathematical model can be observed (Figure 11, Table 4).

The results of the conducted study indicate that the developed voltage–current model enables accurate estimation of the rotor flux of IMs characterized by intense deep-bar effect, in the operating range of the slip frequency. Considering the above, the novel rotor flux estimation scheme can be applied for the rotor-flux-oriented control of the IMs with any rotor construction, including squirrel-cage, double-cage, and solid rotors. Moreover, the elaborated voltage–current model of the rotor flux space vector can be employed as the adjustable model of the Model Reference Adaptive System based estimator for speed-sensorless IM drive applications.

Future work will include experimental studies of the IM rotor-flux-oriented control with the novel rotor flux estimation scheme.

**Author Contributions:** Conceptualization, G.U. and J.R.; Formal analysis, G.U.; Investigation, J.R.; Methodology, G.U. and J.R.; Supervision, A.K.; Validation, G.U. and J.R.; Visualization, G.U.; Writing—original draft, G.U., J.R. and A.K.

**Funding:** This work was supported by the Polish Ministry of Science and Higher Education under research projects: BS/MN-401-312/15 and 03.0.00.00/2.01.01.0001MNSP.E.19.001.

**Conflicts of Interest:** The authors declare no conflict of interest.

## Appendix A

In the presented study, the total apparent electrical power was adopted as base apparent power (input voltampere base).The base values are defined in accordance to the contents of Table A1:

**Table A1.** The per-unit system base values.

| Base Quantity | Symbol | Unit | Formula |
|---|---|---|---|
| Apparent power | $S_b$ | voltampere (V·A) | $3 \cdot U_{1ph} \cdot I_{1ph}$ |
| Frequency | $\omega_b$ | radian per second (rad/s) | $2 \cdot \pi \cdot f_{1N}$ |
| Angular velocity | $\omega_{mb}$ | radian per second (rad/s) | $2 \cdot \pi \cdot f_{1N} \cdot (p_p)^{-1}$ |
| Magnetic flux | $\Psi_b$ | volt second per radian (V·s/rad) | $U_{1ph} \cdot (2 \cdot \pi \cdot f_{1N})^{-1}$ |
| Impedance | $Z_b$ | ohm (Ω) | $U_{1ph} \cdot (I_{1ph})^{-1}$ |
| Inductance | $L_b$ | henry per radian (H/rad) | $U_{1ph} \cdot (I_{1ph} \cdot 2 \cdot \pi \cdot f_{1N})^{-1}$ |
| Torque | $T_b$ | newton meter per radian (N·m/rad) | $3 \cdot U_{1ph} \cdot I_{1ph} \cdot p_p \cdot (2 \cdot \pi \cdot f_{1N})^{-1}$ |

Where $U_{1ph}$ and $I_{1ph}$ are the nominal stator phase voltage and current, respectively, $f_{1N}$ stands for the nominal frequency of stator voltages, and $p_p$ is the number of pole pairs.

**Appendix B**

Figure A1 presents the cross-section dimensions (millimeters) of the tested squirrel-cage (Figure A1a) and solid (Figure A1b) rotors.

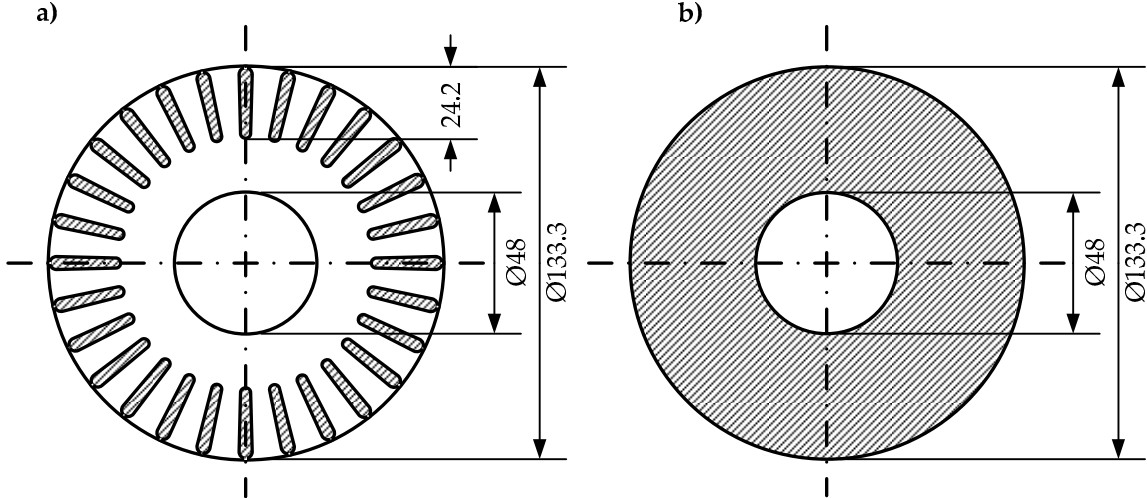

**Figure A1.** The cross-section dimensions of: (**a**) CR-IM and (**b**) SR-IM.

The rated values of the tested IMs corresponding to the adopted machine operating points are included in Table A2. These rated values were determined for the purpose of the presented study and aimed to reduce the breakdown slip of the SR-IM. The rated values were settled so as to maintain approximately equal stator flux amplitudes of the CR-IM and SR-IM, bearing in mind the limitations resulting from rated values of the programmable AC source (AMETEK Model: 3001iX) powering the investigated IMs.

**Table A2.** The rated values of the CR-IM and SR-IM corresponding to the adopted operating points.

| Rating | Unit | CR-IM | SR-IM |
|---|---|---|---|
| Output power | kilowatt (kW) | 2.358 | 1.992 |
| Stator voltage | volt (V) | 400 (wye) | 391 (delta) |
| Stator frequency | hertz (Hz) | 50 | 85 |
| Stator current | ampere (A) | 4.536 | 7.785 |
| Torque | newton meter (N·m) | 15.53 | 9.39 |
| Rotational speed | revolution per minute (r/m) | 1450 | 2030 |
| Power factor | (-) | 0.8819 | 0.6698 |
| Efficiency | (-) | 0.8525 | 0.5641 |
| Stator flux | weber (Wb) | 0.973 | 0.995 |

**Table A3.** The power rating of the individual devices and DC machine employed within the laboratory test bench.

| Name | Description | Unit | Rated Power |
|---|---|---|---|
| AMETEK Model: 3001iX | programmable AC source | kilowatt (kW) | 9.0 |
| DC590P | 4Q thyristor converter | kilowatt (kW) | 7.5 |
| ZSAC4244 – H&H GmbH | programmable electronic load | kilowatt (kW) | 4.2 |
| PCMb 54b | separately excited DC machine | kilowatt (kW) | 6.5 |

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
