# Peer review of "The Novel Rotor Flux Estimation Scheme Based on the Induction Motor Mathematical Model Including Rotor Deep-Bar Effect"

_energies, doi:10.3390/en12142676_

Round 1

Reviewer 1 Report

The problem of deep bar effect is very important in direct drives. It is a good phenomenon to obtain good starting characteristics. However, in this case, the engineer acquires the torque-to-speed characteristics ‘as is’. In contrast, the magnetic field speed with respect to the rotor and the slipping are rather small when we want to provide good characteristics of variable speed drive. Hence, the question is what is the application of your research. This question concerns mainly a squirrel-cage rotor.
As for SR-rotor, the question is why do you use a linear model. In my opinion, it is unacceptable, for the rotor current flows in a thin layer of saturated steel. So, the problem is intrinsically nonlinear.
Also, please, explain, for example, the formula 7a. The time derivative of flux is measured in volts. Then it is multiplied by Tn. So, we have the term measured in volt-second unit, is it so?

Author Response

The authors would like to thank the reviewer for the constructive comments that undoubtedly contributed to the improvement of the paper quality. The paper has been corrected in accordance with the reviewer suggestions. All the revisions in the paper have been highlighted in blue font. 

The authors attach the responses to the reviewer comments. Please see the attachment.

Reviewer 2 Report

·         Very weak research paper, references are not up to date and must be updated, there is to much references from 70s, 80, and 90 and just few from the past 5 years.

·         What are the application of such mathematical model?

·         Please provide a sketch/drawing  of studied rotors design in introduction

·         Introduction part should be improved with paper structure

·         There is no information about tested motor parameters, eg. rated power, current, etc. only simulation results in p.u.

·         What is a purpose of chapter 4 laboratory test bench, if there is no information about power range of devices?

·         None of models compared in Fig6.b corresponded to measurements, please comment

·         Fig 8: looks like SR-IM machine was loaded with lower than rated power, power transferred to the shaft is less than 1. Model is not verified!

·         There is no novelty presented in the paper, all equations/models/results are referenced to another publications.

·         If losses presented in Fig 9 are wrong, that makes flux space vector estimation wrong (or at least has error)

·         What are main advantages and disadvantages of proposed algorithm in comparing to conventional methods 

Author Response

(The authors gave the same response as above.)

Round 2

Reviewer 1 Report

The paper has become much better and can be published.

Reviewer 2 Report

Dear Authors,

thank you for re-submitting your paper and focusing on proper description, now it looks more clear and scientific. However, I have some doubts on application of such machines and control methods, in my opinion synchronous reluctance machines already filling than niche.  Good luck in developing in this field!